# The Role of Mammalian STK38 in DNA Damage Response and Targeting for Radio-Sensitization

**DOI:** 10.3390/cancers15072054

**Published:** 2023-03-30

**Authors:** Takemichi Fukasawa, Atsushi Enomoto, Asako Yoshizaki-Ogawa, Shinichi Sato, Kiyoshi Miyagawa, Ayumi Yoshizaki

**Affiliations:** 1Department of Dermatology, Graduate School of Medicine, The University of Tokyo, 7-3-1 Hongo, Bunkyo-ku, Tokyo 113-8655, Japan; 2Laboratory of Molecular Radiology, Center for Disease Biology and Integrative Medicine, Graduate School of Medicine, The University of Tokyo, 7-3-1 Hongo, Bunkyo-ku, Tokyo 113-8655, Japan

**Keywords:** STK38, DNA damage response, signal transduction, radio-sensitization

## Abstract

**Simple Summary:**

DNA is constantly damaged by internal or external factors. Cells have evolved elaborate damage response mechanisms, namely, DNA damage response (DDR), to preserve genomic integrity. In eukaryotes, kinases play a central role in the DDR. Through a phosphorylation-dependent pathway, these mechanisms quickly transmit a DNA damage signal to the cell cycle checkpoint, cell death, or DNA-repair machinery. Recently, the role of Serine-threonine kinase 38 (STK38) in DNA-damage signaling is emerging. Here, we aim to provide an overview of current topics of STK38 in common mechanisms of regulation, DNA damage signaling, cross-talk between the DDR pathways, cancer, and the potential application for radiotherapy.

**Abstract:**

Protein kinases, found in the nucleus and cytoplasm, play essential roles in a multitude of cellular processes, including cell division, proliferation, apoptosis, and signal transduction. STK38 is a member of the protein kinase A (PKA)/PKG/PKC family implicated in regulating cell division and morphogenesis in yeast and *C. elegans*. However, its function remained largely unknown in mammals. In recent years, advances in research on STK38 and the identification of its substrates has led to a better understanding of its function and role in mammals. This review discusses the structure, expression, and regulation of activity as a kinase, its role in the DNA damage response, cross-talk with other signaling pathways, and its application for radio-sensitization.

## 1. Introduction

Serine/Threonine Kinase 38 (STK38, also known as NDR1: nuclear Dbf2-related 1) is a member of the AGC protein kinase (protein kinase A, G, and C) group. Its homologs have been identified in yeast, *C. elegans*, Drosophila, human, and other organisms [1]. Genetic and cytological analyses using budding and fission yeast or *C. elegans* have reported that STK38/NDR1 homologs Dbf-2 (dumbbell forming protein 2) and Orb6 (origin recognition box 6) are regulators of cell polarity, mitosis, and cell morphology [2,3,4,5]. The function of STK38 in mammalian cells is gradually being understood after being unknown for many years. STK38 is highly abundant in the organs of the immune system [6]. STK38 regulates centrosome duplication and mitotic chromosome alignment [7,8] and is activated by oxidative stress stimuli such as X-ray and hydrogen peroxide. Biochemical or proteomic analyses have shown that STK38 regulates DNA damage responses. In the following sections, we introduce the characteristics and functions of STK38 as a kinase and its application for radio-sensitization.

## 2. Structure, Function, and Regulation of STK38

### 2.1. Structure of STK38

Three structural features characterize the AGC protein kinase family: it consists of two lobes—the N- and C-terminal lobes; the activation segment (AS) on the C-lobe side is adjacent to the ATP-binding site; and the AS is bound to the N-terminal side via an αC helix, with its phosphorylation regulating its conformational change [9]. The groove near the activation site is used for substrate binding, suggesting that the αG helix is involved in substrate binding [10]. Among these, the NDR family, which includes STK38/NDR1, is characterized by the presence of an N-terminal regulatory domain (NTR) and an autoinhibitory sequence (AIS) (Figure 1a,b). It has been shown that S100 calcium-binding protein B (S100B) and Mps one binding (MOB) bind to NTR [11,12]. STK38 has a unique basic-residue-rich insert (residues 244–276) in the catalytic domain and gets autoinhibited by it. The interaction between MOB with the NTR domain of STK38 releases the auto-inhibition, permitting the efficient autophosphorylation of STK38 on S281, which is in the activation segment, and thus activates STK38 activity [12]. The 3D structure in Figure 1b clearly depicts how the autoinhibitory sequence covers the autophosphorylation site of S281. In addition, the autophosphorylation site of S281 and the hydrophobic motif (HM) T444 are phosphorylation sites that positively regulate STK38 [13]. In the NTR, in addition to the regulatory factor binding domain, phosphorylation S6, T7, S10, S11, and T74 have been identified [14,15]. GSK-3 phosphorylates STK38 at S6 and T7 and inhibits STK38’s full activation. Experiments with a phosphorylation-defective mutant demonstrated that phosphorylation of S91 is essential for STK38 stability, preventing the degradation of the enzyme by the calpain pathway [16]. K118 is involved in ATP binding. Interestingly, somatic mutations in *STK38* are known in normal and tumor tissues of cancer patients. Several modifications are known, especially in lung, ovarian, and skin tumors (Figure 1a). These mutations span regulatory and structural domains such as NTR, αC helix, AIS, and αG helix, which may affect STK38’s interaction with other molecules and kinase activity.

### 2.2. Regulation of Expression and Activity of STK38

Murine *Stk38* mRNA was expressed in the spleen, lung, thymus, heart, brain, pancreas, muscle, and fatty tissue [21]. Deletion analysis and site-directed mutagenesis experiments demonstrated that Sp1 (specificity protein 1), a well-known member of a family of transcription factors which are implicated in an ample variety of essential biological processes, was required for the *STK38* promoter activity [22]. Protein expression analysis revealed a high expression of STK38 in lymphoid tissues such as the thymus, spleen, and lymph nodes [17]. The subcellular localization of STK38 was originally thought to be nuclear with a nuclear localization signal (NLS)-like the sequence in the insertion sequence between subdomains VII and VIII. However, STK38 is present in the cytoplasm and the nucleus [1,23,24,25]. Recently, several mechanisms have been elucidated regarding the protein stability of STK38. SOCS2 (The suppressor of cytokine signaling 2) interacts with STK38 and promotes its degradation through K48-linked ubiquitination [26]. MEKK2 phosphorylates S91 of STK38, and STK38 is susceptible to degradation by the calpain pathway unless the S91 residue is phosphorylated [16]. On the other hand, STK38L (STK38-like, also known as NDR2, has 86% homology at the amino acid level with STK38) had a different distribution. STK38L is expressed predominantly in the brain [10,24], heart, thymus [10], small intestine, stomach, and testis [27]. Subcellular localization of STK38L is in the cytoplasm and on the surface of the plasma membrane [10,21,23,27,28].

Various stress stimuli to STK38 resulted in little change in activity with etoposide, anisomycin, and sorbitol. At the same time, treatment with hydrogen peroxide or X-ray significantly increased STK38 activity [15,29]. On the other hand, okadaic acid, a putative inhibitor of protein phosphatase type 2A (PP2A), inhibits S281 and T444 dephosphorylation and positively activates STK38 in vitro [13]. Mammalian STE20-like kinase 3 (MST3) was identified as an upstream molecule that regulated STK38 by phosphorylation of the hydrophobic motif site [14,28,29] (Figure 2). We also found that glycogen synthase kinase 3 (GSK-3) phosphorylates S6 and T7 of STK38 and negatively regulates its activity by primarily leveraging protein kinase A (PKA)-mediated priming phosphorylation on residues S10 and S11 (Figure 2) [15]. In addition to the regulation by phosphorylation as described above, its kinase activity is also regulated by STK38 interacting factors via the NTR. STK38 has an insert in the catalytic domain. The insert sequence contains many positively charged amino acids and has an autoinhibitory function. Binding of MOB1 to STK38 induces the release of the autoinhibition caused by the autoinhibitory sequence, stimulating STK38 activity [12]. While the binding of MOB1 to STK38 increases STK38 activation, the binding of MOB2 to STK38 results in the inhibition of STK38 activation, highlighting a divergent effect of MOB isoforms binding to STK38. It is also known that STK38 activity is constitutively activated when STK38 colocalizes at the plasma membrane with human MOBs [28]. STK38 is activated through a direct interaction with EF-hand Ca^2+^-binding proteins of the S100 family. S100B binds to the NTR and increases STK38 kinase activity in a Ca^2+^ concentration-dependent manner [14].

## 3. Involvement of STK38 in DNA Damage Signaling

Ionizing radiation (IR) produces a wide array of DNA lesions, including DNA base damage (base modification), single-strand breaks (SSBs), double-strand breaks (DSBs), sugar damage, DNA-DNA cross-links, and DNA-protein cross-links [30]. Of the many types of DNA damage in the cell, DSBs are the most dangerous, their repair being intrinsically more complex than that of other types. Cells have evolved elaborate damage response mechanisms to cope with constant attacks on their DNA and to maintain genomic stability. These mechanisms quickly transmit a DNA damage signal to the checkpoint arrest, apoptotic, or DNA-repair machinery. DNA damage signal transduction is mediated by the phosphatidylinositol-3-kinase-related protein kinase (PIKK) family in mammalian cells [31]. DNA damage is recognized by sensor proteins such as the MRE11-RAD50-NBS1 (MRN) complex. MRN complex recruits one of the PIKK family members, ataxia telangiectasia mutated (ATM) to the DSB sites, leading to a conformational change of the ATM protein [32]. This causes autophosphorylation and activation of ATM, leading to initiation of the DNA damage signaling cascade. Ufmylation is a ubiquitin-like post-translational modification that attaches the ubiquitin-fold modifier 1 (UFM1) to target proteins. This process regulates several cellular functions, including DNA damage response. UFM1 promotes ATM activation, amplified by mono-ufmylation of the K31 residue of histone H4 by UFM1-specific ligase 1 (UFL1) [33]. STK38 is a reader for histone H4 ufmylation to promotes ATM activation in a kinase-independent manner [34]. STK38 has a UFM1 binding motif and binds to ufmylated H4, recruiting histone-lysine N-methyltransferase SUV39H1 to the DSB site, resulting in trimethylation of H3K9 and activation of Tip60, leading to ATM activation (Figure 3).

DNA-PK (DNA-dependent protein kinase), another member of the PIKK family, plays a vital role in DSB repair. DNA-PK activity is required for non-homologous end joining, a simpler and more error-prone DNA repair mechanism that rejoins the two severed DNA ends in a sequence-independent fashion [35]. DNA-PK has been reported to be associated with STK38. During starvation, the cells became resistant to ionizing radiation with increased activity of DNA-PK or STK38. In particular, the knockdown of DNA-PKcs suppressed starvation-induced phosphorylation of the STK38 activation segment site, S281 [36]. Moreover, LATS1 (large tumor suppressor kinase 1) and STK38L, which belong to the NDR family, have a history of being identified by a wide screen of motif SQ/TQ sequences phosphorylated by DNA injury [37]. These reports indicate STK38 and STK38L may function downstream of the ATM, ATR (ataxia telangiectasia and RAD3 related), and DNA-PK.

RAD51 is a central player in homologous recombination (HR)-dependent DNA repair, using an undamaged sister homolog as a template, thereby providing a process capable of achieving high fidelity even if sequence information is lost at the break site [38]. RAD51 is loaded onto single-stranded DNA, catalyzes the search for homologous sequences, and promotes strand invasion. A recent report demonstrated that loss of hMOB2, known as a regulator of STK38, disrupts HR-dependent DSB repair by interfering with the phosphorylation and accumulation of the RAD51 recombinase on resected single-strand DNA (ssDNA) overhangs, leading to sensitization of cancer cells to DSB-inducing drugs [39]. Interestingly, hMOB2 deficiency sensitised cancer cells towards PARP inhibitors such as olaparib, rucaparib and veliparib, and enhances the radio-sensitising effect of olaparib treatment. STK38 is also involved in nucleotide excision repair (NER). UV radiation induces dipyrimidine photoproducts, mainly cyclobutyl-pyrimidine photodimers—CPDs—and some pyrimidine (6-4) pyrimidone photoproducts (6-4PPs) in cellular DNA through linkage of two adjacent pyrimidine bases upon photoexcitation. NER is the primary pathway that removes the helix-distorting DNA strand damage induced by ultraviolet (UV) irradiation [40]. The xeroderma pigmentosum (XP) A–G proteins are the core factors involved in the NER pathway. The NER system consists of a sequential and coordinated process requiring the XP proteins, including lesion recognition and verification and dual incision followed by DNA re-synthesis and ligation. STK38 accumulated in the nucleus after UV-irradiatin and interacts with a NER factor XPA which functions in damage-verification and assembly of NER incision complexes, and depletion of STK38 delays the repair of UV-induced cyclobutane pyrimidine dimers with modulating ATR signaling [41].

As described above, STK38 might be involved in various DNA damage signaling pathways.

## 4. Role of STK38 in DNA Damage-Induced Cell Cycle Checkpoint

Cell cycle checkpoint arrest after DNA damage has two potential effects: it allows additional time for DNA repair before cell cycle progression, and it can permanently prevent the proliferation of severely damaged cells [42]. In response to DSBs, checkpoint machineries are activated, resulting in cell-cycle arrest at DNA-damage response checkpoints. CDC25A plays a vital role in the G1/S and G2/M transition by dephosphorylating and activating cyclin/cyclin-dependent kinase (CDK) complexes [43,44]. CDC25A itself is known to be degraded in response to DNA damage. For example, in the presence of DSBs caused by IR, ATM phosphorylates checkpoint effector kinase CHK2, and then the activated CHK2 phosphorylates S123 of CDC25A, targeting the phosphatase for ubiquitin-dependent proteasomal degradation [43] (Figure 3). STK38 phosphorylates S76 of CDC25A, and the S76A mutant resisted IR-induced degradation. Interestingly, the depletion of STK38 also did not cause DNA damage-induced CDC25A degradation and subsequent G2 arrest [45]. These results suggest that in response to DNA damage, direct phosphorylation of S76 of CDC25A by STK38 triggers subsequent degradation of CDC25A and activation of G2/M checkpoints.

ATR activation occurs in response to replication stress and during normal DNA replication and mitosis [31]. CHK1 activated by ATR phosphorylates S76 of CDC25A [44]. STK38 is activated by oxidative stresses, including X-rays and hydrogen per oxidate and phosphorylates S76 of CDC25A. Therefore, CDC25A is phosphorylated at the same and different sites depending on the type of stimulus. On the other hand, it has been shown that Cyclin D1 enhances STK38/STK38L activity to promote G1/S transition independently of CDK4 [46]. Furthermore, STK38 and STK38L are activated in G1 by MST-3 and control G1/S cell cycle transition by phosphorylating p21 (cyclin-dependent kinase inhibitor 1A; CDKN1A). The depletion of STK38 and STK38L results in G1 arrest and subsequent significant decrease in cellular proliferation [47]. Therefore, depending on the cell cycle and the type of stimulus, STK38 is activated and regulates the cell cycle by phosphorylating common or different substrates.

## 5. Role of STK38 in Cell Proliferation, Cell Survival, and Autophagy

The oncoprotein MYC is a transcription factor that regulates critical cellular processes including cell growth, differentiation, metabolism, and apoptosis, and is frequently dysregulated in many human cancers [48]. Regulatory network analysis in human B-cells demonstrated that STK38 regulates MYC transcriptional activity and its protein stability in a kinase activity-dependent manner [49]. STK38 silencing significantly decreases MYC levels and increases apoptosis. It has been shown that STK38 potentiates nuclear factor-κB (NF-κB) activation by its kinase activity [50]. NF-κB, an inducible transcription factor that is essential for inflammatory responses, and plays a critical role in regulating the survival, activation and differentiation of innate immune cells and inflammatory T cells [51]. Aberrant NF-κB activity plays a critical role in tumorigenesis and acquired resistance to conventional cancer treatments [52]. The overexpression of STK38 amplified TNF-α induced NF-κB signaling and increased clonogenicity, suggesting the possible role of STK38 as a driver of oncogenic growth in NF-κB-activated tumors [26]. On the other hand, STK38 has been demonstrated to bind to several MAP kinase kinase kinases (MAP3Ks) members, such as MEKK1 and MEKK2, located upstream of the stress-activated protein kinase (SAPK) signaling cascade. STK38 functions as a negative MAP3K regulator by inhibiting their autophosphorylation and suppressing the downstream signaling [53]. GSK-3 limits STK38 activity, and STK38 is released from GSK-3 mediated negative regulation through the action of AKT and activated in response to oxidative stress [15]. The knockdown of STK38 enhanced oxidative-stress-induced apoptosis with an increase in c-Jun N-terminal kinase (JNK) phosphorylation, suggesting that STK38 protects cells from oxidative stresses through inhibiting SAPK signaling or playing an anti-apoptotic function downstream of AKT.

Autophagy is a conserved and highly regulated process of the lysosomal pathway that cleanses cells by recycling damaged proteins, macromolecules, and organelles [54]. Autophagy acts as a pro-survival or pro-death mechanism, depending on cell types, context, and stimulus. Several reports have demonstrated that autophagy is induced by radiation [55]. Cells deficient in autophagy have been shown to accumulate higher levels of mutated DNA with enhanced DNA damage and less efficient damage repair, indicating that autophagy functions to maintain genome stability [56,57]. STK38 supports the interaction of the exocyst component Exo84 with Beclin1 and RalB, which is required to initiate autophagosome formation [58]. Dual loss of *Stk38/Stk38l* in neurons causes neurodegeneration with decreased autophagy [59]. The levels of LC3-positive autophagosomes were reduced in the *Stk38/Stk38l* knockout neurons, and then reduced autophagy mediates accumulation of transferrin receptor p62 and ubiquitinated proteins and neurotoxicity. Thus, STK38 is possibly involved in the induction of DNA-damage-induced autophagy.

## 6. Involvement of STK38 in Malignancy

The *STK38* mRNA expression levels have been reported as up-regulated in breast [60], ovarian [61], and lung [62] cancers. The *Stk38* knockout mice are typically born. Tissue expression analysis showed that an increase in STK38L protein functionally compensated for the loss of STK38. However, the knockout mice are susceptible to developing T-cell lymphomas such as angioimmunoblastic T-cell lymphoma (AITL) in old age [17]. It has already been reported that STK38 overexpression leads to centrosome overduplication [7], leading to chromosomal instability. *STK38* knockdown suppresses the growth of MYC-addicted tumors in vivo, thus providing a novel candidate target for treating these malignancies [49]. STK38 is involved in malignant tumor invasion and metastasis, cancer stem cells, and even development. For example, STK38 increases NOTCH1 (notch receptor 1) signaling activity by impairing Fbw7 (F-box and WD repeat domain-containing 7) mediated the degradation of NICD (notch intracellular domain) to enhance breast cancer stem cell properties [63]. Moreover, STK38 is required for anchorage-independent soft agar growth, and in vivo xenograft growth of Ras-transformed human cells, in which STK38 supports the Ras-induced transformation by promoting detachment-induced autophagy [64]. The expression level of inwardly rectifying potassium channel 2.1 (Kir2.1) is elevated in invasive gastric cancer cells. Kir2.1 relates to properties such as invasiveness and metastasis, and epithelial-mesenchymal transition. These properties are reported to result not from the function of Kir2.1 as an ion channel but from its interaction with STK38 to inhibit ubiquitination and degradation of MEKK2, thereby enhancing MEKK2-MEK1/2-ERK1/2 (extra-cellular signal-regulated kinase 1/2) signaling cascade [65] (Figure 2). Recently, a pan-cancer analysis of *STK38* reveals that the expression of *STK38* is associated with patient survival, immune cell infiltration such as NK cells, cancer-associated fibroblasts, tumor mutation burden and microsatellite instability in several human cancers [66]. However, there is also evidence that STK38 can reduce the cell proliferation of glioblastomas and inhibit the metastasis of prostate cancer [67]. STK38/STK38L kinases phosphorylate the transcriptional coactivator yes-associated protein (YAP1) and thereby negatively regulate YAP1 transcriptional activity, suggesting that STK38/STK38L function as tumor suppressors [68]. STK38 kinase might play opposing roles in tumors by serving as a tumor suppressor or an oncogenic factor in different conditions. On the other hand, STK38L expression is elevated in a subset of primary PDACs (Pancreatic ductal adenocarcinomas) and PDAC cell lines displaying the aberrantly differentiated endocrine exocrine (ADEX) subtype characteristics including overexpression of mutant KRAS, and depletion of STK38L in a subset of ADEX subtype cell lines inhibits cell proliferation and induces apoptosis [69]. High *STK38L* mRNA expression is associated with decreased overall patient survival in PDACs. STK38L contributes cell invasion and cytokinetic abnormalities of RASSF1A (Ras association (RalGDS/AF-6) domain family member 1)-depleted lung cancer cells by activating YAP [70].

## 7. Targeting of STK38 for Radio-Sensitization

STK38 plays multi functions in DNA-damage responses and then is suggested to be a potent target for radio-sensitization. HSP90 dynamically promotes the conformational maturation of its client proteins and protects them from degradation by assembling client-HSP90 complexes using the chaperone machinery [71]. 17-allylamino-17-demethoxygeldanamycin (17-AAG), an HSP90 inhibitor, is a molecularly targeted agent and combat tumors by inhibiting HSP90’s intrinsic ATPase activity, causing HSP90’s client proteins to be degraded via the ubiquitin-proteasome pathway [72]. In tumor cells, HSP90 is present in multi-chaperone complexes with high ATPase activity and a solid binding affinity for 17-AAG. Since this is not the case in normal cells, 17-AAG is selective for tumors. STK38 was demonstrated to interact with HSP90, and the interaction was dissociated by the HSP90 inhibitor ganetespib [73]. Interestingly, 17-AAG markedly suppressed the expression of STK38 not only at the protein level but also at the transcriptional level and reduced its kinase activity [22]. Knockdown of *STK38* also induced radio-sensitization as well as 17-AAG. The suppression of *STK38* expression by 17-AAG may inhibit the DNA damage response such as DNA repair or cell cycle arrest, leading to a part of the 17-AAG-mediated radiosensitization. These findings suggest that STK38 is a potent and unique target for 17-AAG-mediated radio-sensitization.

Recent reports have been published on the control of STK38 by temperature. Hyperthermia increases the cell temperature and induces many biochemical changes, such as the protein degradation and an increased intracellular calcium ion concentration [74]. Heat-induced protein denaturation, aggregation, or degradation is a key event in the disruption of cellular homeostasis [75]. Hyperthermia has long been known to enhance the anti-tumor and therapeutic effects of radiation. Proteomic analysis identified STK38 as a factor whose expression is decreased in a heat-specific manner [16]. The calpain inhibitor calpeptin suppressed hyperthermia- or calcium ionophore A23187-induced degradation of STK38. In vitro cleavage assay demonstrated that calpain I directly cleaves STK38 at the proximal N-terminal region. We further showed that MEKK2 prevented both heat- and calpain-induced cleavage of STK38 through direct phosphorylation at S91 (Figure 2). Hyperthermia also decreased the expression level and activity of MEKK2 [16,76]. Thus, the inactivation of MEKK2 by hyperthermia suppresses STK38 phosphorylation, making it susceptible to thermally induced degradation. Several reports demonstrated that silencing *STK38* leads to decreased clonogenicity [22,47]. These findings suggest that STK38 may be one of the factors responsible for the antitumor effect of 17-AAG and hyperthermia. Since STK38 regulates DNA damage responses and malignancy, STK38-targeted therapies may have synergistic effects in radiotherapy.

## 8. Summary and Conclusions

STK38 is activated by oxidative stresses such as X-ray irradiation and plays a vital role in regulating DNA-damage responses such as DNA repair, cell cycle checkpoint, autophagy, and apoptosis. The expression of *STK38* was up-regulated in several cancers and associated with cancer immunity. STK38 is involved in malignant tumor invasion and metastasis, and its overexpression leads to chromosomal instability. The HSP90 inhibitor 17-AAG sensitized cells to X-ray through down-regulation of *STK38*. Moreover, thermal stress decreases STK38 stability via a negative impact on MEKK2 and raises the potential of STK38 as a target in hyperthermia. Cells with reduced STK38 expression decrease cell proliferation, clonogenicity, and are more vulnerable to X-rays or oxidative stress apoptosis. Thus, STK38-targeted therapies may have synergistic effects in radiotherapy. Another therapeutic or radio-sensitizing impact may also be achieved by an inhibitor specific to STK38, which is awaiting development.

STK38 has been reported to interact with various molecules and play multiple roles at the cellular and individual levels. Research surrounding STK38 is currently being actively conducted at different facilities. Further analysis of STK38 on DNA damage signaling and response may be helpful in development of the new radio-sensitizers.

## Figures and Tables

**Figure 1 cancers-15-02054-f001:**
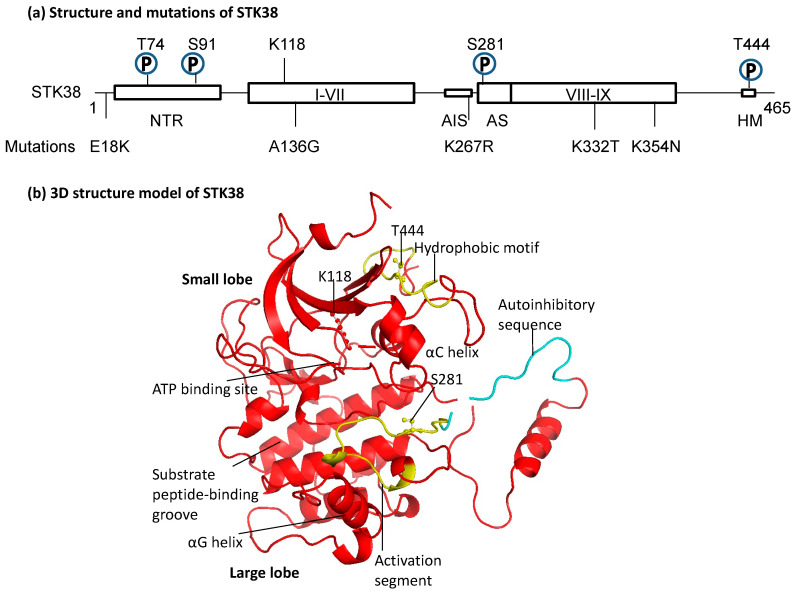
Most prominent three-dimensional structural features of STK38. (**a**) Domain structure of STK38. NTR at the N-terminus, subdomains I-VII, VIII-IX, activation segment (AS; aa277–292), and a hydrophobic motif (HM, aa439–451) at the C-terminus. Phosphorylation is denoted by P. The active center, a lysine residue, is located in subdomain II, and the autoinhibitory sequence (AIS, aa265–276) is situated between subdomains VII and VIII. Major somatic high-frequency mutations of STK38 in each cancer tissue are also described. Mutations in E18K are found in skin tumors, A136G in lung cancer, K332T and K354N in ovarian cancer, and K267R in normal tissues of carcinoma patients [17]. (**b**) Prediction of the domain and steric structure of STK38. Small lobe, large lobe, αC helix, and αG helix are shown in addition to the domain of a, and the three phosphorylation sites are indicated by ball-and-stick. Only the main part of the tertiary structure of STK38 is shown (Lys80-Ala456). Web site (Automated comparative protein-modeling server, Swiss Model (http://swissmodel.expasy.org/ (accessed on 31 March 2021)) was used to predict the steric structure [18,19,20].

**Figure 2 cancers-15-02054-f002:**
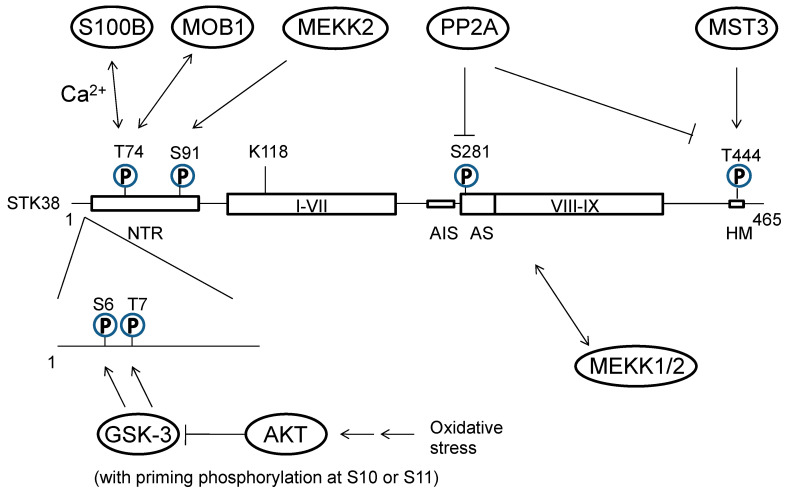
The regulation of STK38 by phosphorylation and protein–protein interactions. The regulatory region (NTR) at the N terminal of STK38 binds positive regulators such as MOB and S100B, which promote autophosphorylation (S281) and activation of STK38. Phosphorylatin of S6 and T7 at the STK38 by GSK-3 reduces STK38 activity. Upon activation of AKT (protein kinase B, PKB) by oxidative stress, GSK-3 is inactivated, which releases the negative regulation of STK38, resulting its activation. MEKKs (MEK kinases), which play essential roles in MAPK (mitogen-activated protein kinase) cascade and the oxidative stress-induced response, are negatively regulated by STK38 binding. MST-3 phosphorylates T-444 at the STK38 C-terminus and positively regulates STK38. Phosphorylation of S91 by MEKK2 is critical for STK38 stability, as the enzyme is susceptible to degradation by the calpain pathway, only to be prevented by the phosphorylation of this residue.

**Figure 3 cancers-15-02054-f003:**
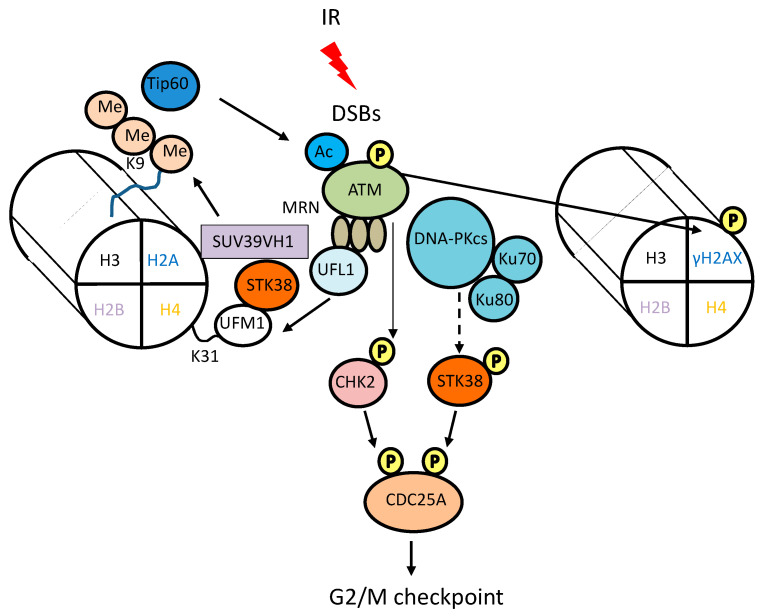
Involvement of STK38 in DNA damaging signaling. IR produces many DNA lesions, including DNA double-strand breaks (DSBs). DNA damage is recognized by sensor proteins such as the MRN complex. MRN complex recruits the signal-transducing kinase ATM and collaborates for its activation. UFM1-specific ligase 1 (UFL1) is recruited by the MRN complex and mono-ufmylates of the K31 residue of histone H4. STK38 has a UFM1 binding motif and binds to ufmylated H4, recruiting SUV39H1 to the DSB site, resulting in trimethylation of H3K9. The acetyltransferase Tip60 is activated in a histone H3K9me3-dependent manner and acetylates ATM, promoting to ATM activation [34]. Activated CHK2 by ATM phosphorylate S123 of CDC25A, accelerating its degradation through the ubiquitin–proteasome pathway, thus leading to G2 arrest. DNA-PK plays a vital role in the repair of DSBs. Knockdown of DNA-PKcs suppresses phosphorylation of STK38 activation segment site, S281. Activated STK38 by IR phosphorylates S76 of CDC25A, leading to subsequent degradation of CDC25A and activation of G2/M checkpoint. Me and Ac indicate methylation and acetylation, respectively.

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
