# Peer review of "The Role of Mammalian STK38 in DNA Damage Response and Targeting for Radio-Sensitization"

_cancers, 2023, doi:10.3390/cancers15072054_

Round 1

Reviewer 1 Report

The review article by Fukasawa et al provides an overview of STK38 kinase, implicated in DNA-damage signaling and malignancy, and its potential application in radio-sensitization of tumor cells. The subsections of the review are appropriately organized and extensively cited. However, the organization of the content within the subsections could benefit from proofreading. 

It will be helpful to the readers if the authors discuss the implications of certain results where contradictory findings have been reported regarding STK38’s role in cell-cycle progression, DNA-damage signaling, and malignancy.

1. For example, line 177-180: even though STK38 directly phosphorylates Ser-76 of CDC25A and primes it for degradation, this functions apparently is not attributed in DNA damage-induced CDC25A degradation.

2. Line 196-200: STK38 is required for G1/S transition but then it is also a target for DNA-damage induced checkpoints. The authors should discuss these different cell cycle roles of STK38.

Reviewer 2 Report

The review by Fukasawa and co-authors describes some of the current knowledge around the cellular function of the protein kinase STK38 (NDR1), focusing on its potential role(s) in regulation of the DNA damage response and proposing that targeting p38 function might be an appropriate mechanism for radio-sensitization of tumors, thus increasing the effectiveness of radiotherapy for treating cancer.

In this respect, I think this review has the potential to be of interest to researchers across a range of areas and not just those interested in DNA damage responses, checkpoints and repair. However, the manuscript in its current form is not particularly comprehensive in scope and its coverage of STK38’s proposed functions in the DNA damage response is overly simplistic. The main focus of the review is to propose targeting STK38 as a way of radio-sensitizing but some evidence from work from the authors published some time ago is provided the discussion of ways that this sensitization may be carried  out is relatively superficial.

Main comments:

1. The contribution of STK38 to the DNA damage response is discussed largely on the basis that p38 is involved in ATM activation and may be a putative substrate of DNA-PK/ATM due to its appearance in screens for damage dependent SQ-TQ phosphorylation. How does the apparent role of NDR1 in the NER pathway and proposed regulation of ATR activity (Park et al BBRC 2015) fit with the model proposed?

2. The phosphorylation of CDC25A on Ser76 to induce checkpoint-dependent degradation is also controversial. While there may be a contribution from STK38 (either direct or indirect), there is uncertainty about the contribution of the checkpoint kinases Chk1 (via ATR) and Chk2 (ATM/DNA-PK) (Jin et al JBC 2008) to making this modification. How do the authors reconcile this with their model?

3. The authors cite their own evidence that inhibition of HSP90 via 17-AAG suppresses STK38 expression and increased radiosensitivity of cells, along with a putative role in the DNA damage response, there is relatively little discussion, given that this appears to be the objective of the review, to discussion of how this might be effected (other than a STK38 inhibitor). Why might this be more (or less) effective than targeting of other components of the DDR (some of which are already being explored)? 

The manuscript is also let down by the writing style. There is an element of this caused by some issues around English language but the structure is relatively poor in that there are sections where it feels that the information is presented as a series of brief, seemingly isolated sentences which makes it difficult to follow.

The manuscript is also let down by the writing style. There is an element of this caused by issues around English language but the structure is relatively poor in that there are sections where it feels that the information is presented as a series of brief, seemingly isolated sentences

Reviewer 3 Report

This is rather a mini-review on the role of STK38 in DDR. The Sections of the MS are logically organized, accompanied by 62 references. The Figures are well-designed and not overloaded. The subject on STK38 is apparently a growing field and the information on the role of STK38 in humans seems to be quite scarce, thus, it is important to compile the available data. Even if the paper is in a solid shape, it suffers from some shortcomings. To improve the MS, the authors should put more effort in explanation of some events/ mechanisms in more detail. In addition, generally seen, the English language editing should be carefully conducted.

- The title should include the word ‘human’ or ‘mammalian’ STK38 from the following reasons: a) the kinase is not well known among scientists (in comparison to e.g., SAPK/ MAPK), b) the authors are not really reporting anything on STK38 of e.g., S. cerevisiae/S. pombe, C. elegans, or D. melanogaster.

- The authors should use the same nomenclature for genes and proteins throughout the text; for human genes it is a standard to use big letters in italic, for human proteins big letters in a regular style; for murine genes small letters. Also, for amino acids they should uniquely use the three-letter-code, e.g., Ser-77, Thr-144, instead of S77, T144, etc. This should not be done when the amino acids are depicted as a part of a protein, e.g., H3K9, or in case of mutations.

- The English language should be used in a clear, elegant, and logical way; some sentences are ‘clumsy’ written, because they were just taken from the context of other published papers (references); this often leads to misunderstanding (semantic problems), e.g., which factor is actually responsible for which action. The authors should use either the American or the British version of English. The spelling and grammar (particularly usage of the simple present tense in singular versus plural) should be checked.

- The authors should not use auto-plagiarism (within the same MS text or elsewhere); e.g., the same sentences as in the Simple Summary can be found in the Section 3 (lines 138-140); the authors should rewrite the Simple Summary.

- The authors mentioned DNA-PK (non-homologous end joining, NHEJ) to be linked with STK38, and wrote NHEJ ‘is a simpler and more error-prone DNA repair mechanism’ (line 156). Thus, they are obviously comparing it with something. I would guess, with homologous recombination (HR) as a mechanism of DSB repair. If that’s so, then, they should state this, and also comment on a putative connection between STK38 and HR (which also takes place in S and G2). This would be a valuable information for a broader readership.

- Is it something known about regulation of CDC25C by STK38 upon DNA damage induced by IR, or other DNA damage inducing compounds? Inactivation of CDC25C is also involved in induction of the G2/M arrest.

- The authors should describe the process of ufmylation, since this ubiquitin-like pathway is not mentioned that often in the literature, as e.g., the classical ubiquitination/sumoylation.

- The authors should rewrite the title and the subtitle of the Section 7. Instead of ‘its application (title) or application of STK38 (Section 7), ‘targeting (of) STK38 for radio-sensitization’ would be more appropriate.

- As I understood from the last sentence of the Conclusion Section, there are no STK38 small-molecule inhibitors in a pre-clinical development? Is this right?

- Line 91: replace nuclear migration signal with ‘nuclear localization signal (NLS)-like sequence’

- If the corresponding author(s) did not receive any external funding for this paper, then under Author Contributions, ‘funding acquisition’ is dispensable.

Round 2

Reviewer 2 Report

The changes you have made to the organization, style and English make the manuscript much improved

Reviewer 3 Report

The MS has been substantially improved. There are still some minor spelling mistakes, or inconsistence in denominating aminoacids, genes or proteins (STK38 vs. stk38; T44 vs. Thr44) but the style has been considerably improved.